# Impact of alexithymia, speech problems and parental emotion recognition on internalizing and externalizing problems in preschoolers

**Irina Jarvers**[1]*, **Eva Kormann**[2], **Daniel Schleicher**[1], **Angelika Ecker**[1], **Stephanie Kandsperger**[1], **Romuald Brunner**[1]

**1** Department of Child and Adolescent Psychiatry and Psychotherapy, University of Regensburg, Regensburg, Germany, **2** Institute of Interactive Systems and Data Science, Graz University of Technology, Graz, Austria

* irina.jarvers@ukr.de

## Abstract

### Background

Alexithymia, characterized by difficulty identifying and describing emotions and an externally oriented thinking style, is a personality trait linked to various mental health issues. Despite its recognized importance, research on alexithymia in early childhood is sparse. This study addresses this gap by investigating alexithymia in preschool-aged children and its correlation with psychopathology, along with parental alexithymia.

### Methods

Data were analyzed from 174 parents of preschoolers aged 3 to 6, including 27 children in an interdisciplinary intervention program, all of whom attended regular preschools. Parents filled out online questionnaires assessing their children's alexithymia (Perth Alexithymia Questionnaire–Parent Report) and psychopathology (Strengths and Difficulties Questionnaire), as well as their own alexithymia (Perth Alexithymia Questionnaire) and emotion recognition (Reading Mind in the Eyes Test). Linear multivariable regressions were computed to predict child psychopathology based on both child and parental alexithymia.

### Results

Preschool children's alexithymia could be predicted by their parents' alexithymia and parents' emotion recognition skills. Internalizing symptomatology could be predicted by overall child alexithymia, whereas externalizing symptomatology was predicted by difficulties describing negative feelings only. Parental alexithymia was linked to both child alexithymia and psychopathology.

### Conclusions

The findings provide first evidence of the importance of alexithymia as a possible risk factor in early childhood and contribute to understanding the presentation and role of alexithymia.

**Data Availability Statement:** The data that support the findings of this study are openly available from the Open Science Framework (OSF) database at

https://osf.io/kzgjn/?view_only=
56619072bcb642dc898ad4438dafe7ac. A doi will
be provided once the data has been set to public
after peer-review.

**Funding:** The author(s) received no specific
funding for this work.

**Competing interests:** The authors have declared
that no competing interests exist.

This could inform future research aimed at investigating the causes, prevention, and intervention strategies for psychopathology in children.

## Introduction

Mental health issues represent a significant global concern, ranking among the leading causes of mortality worldwide [1]. Among affected demographics, children and adolescents stand out as particularly vulnerable, experiencing mental health disorders at alarming rates, with prevalence rates reaching up to 13.40% [2]. Recent research indicates an increasing prevalence in this population [3], exacerbated by the COVID-19 pandemic. Longitudinal studies consistently highlight the critical link between emotional and behavioral challenges in early childhood and the manifestation of mental health difficulties in adolescence [4]. These adolescent mental health issues often persist into adulthood [5], underscoring the enduring impact of early-life struggles on long-term mental well-being [6]. Given the predictive nature of early childhood mental health challenges and their widespread occurrence, it is imperative to examine potential risk factors. Understanding these factors is essential for informing prevention and intervention strategies aimed at mitigating the burden of mental health disorders across the lifespan.

In recent years, alexithymia—characterized by difficulties in identifying and describing one's own emotions, coupled with an externally oriented thinking style [7, 8]—has garnered increasing attention as a risk factor influencing mental health outcomes [9]. Alexithymia profoundly impacts various psychiatric disorders during both adolescence and adulthood, including eating disorders [10], depression [11], and substance abuse [12]. Longitudinal investigations have underscored alexithymia's role as a broad-spectrum risk factor for overall psychopathology in adolescence [13].

While much of the existing research has focused on adults or school-aged children who can self-report their alexithymia levels [13, 14], it is imperative to extend this inquiry to early childhood. Several theories attempt to explain the relationship between alexithymia and psychopathology. The vulnerability hypothesis posits that alexithymia constitutes a risk factor for developing of mental health problems [11], while the reactivity hypothesis postulates that alexithymia is a consequence of psychopathology [11]. To date, few studies involving preschool children have investigated parental alexithymia [15–17], identifying negative correlations between parental alexithymia and preschool children's emotional skills [15] and positive correlations with psychopathology in toddlers [16]. However, no scientific studies have examined alexithymia in preschool children and its associations with psychopathology. Identifying alexithymia and its potential links to psychopathology in preschool children, before mental health conditions have fully manifested, might provide additional support for the vulnerability hypothesis.

During early childhood, caregivers (mainly parents) and teachers serve as primary informants regarding children's mental health and alexithymia, given young children's limited linguistic and introspective capacities for self-reporting [14]. However, it is essential to acknowledge that caregiver reports may be influenced by factors such as the caregivers' own challenges with emotion recognition [18] or their levels of alexithymia [15–17]. Caregivers' struggles to identify and interpret their child's emotions, particularly if they also experience difficulties in recognizing and expressing their own emotions, can compromise the accurate assessment of alexithymia in the child. Previous research indicates that alexithymia in parents

is linked to diminished emotional skills in their children and can lead to underestimation of the child's emotional abilities. [15]. This suggests that parents with higher levels of alexithymia may provide less precise assessments of their child's emotional skills, even when their children already exhibit reduced emotional performance. Although parental assessments of psychopathology in preschool children have demonstrated high reliability [19, 20], the presence of alexithymia and variations in emotion recognition skills among parents may influence their evaluations of their children's emotional skills. To mitigate this issue, it is essential to assess parental alexithymia and emotion recognition skills systematically and include them as covariates when investigating parent-reported child alexithymia.

Our study aims to fill the gap of alexithymia research in preschool children by conducting an online survey for parents, assessing parent-reported alexithymia levels and psychopathology in children aged 3 to 6 years. Through this investigation, we seek to elucidate the early emergence of alexithymia as a potential precursor to later mental health challenges, thereby informing targeted interventions and prevention strategies. Additionally, our study aims to investigate the influence of parents' emotion recognition skills and alexithymia on children's alexithymia and psychopathology. To achieve this, we queried parents on their levels of alexithymia and emotion recognition skills, their children's participation in early interdisciplinary intervention programs (IIP), and demographic details. IIPs cater to children from birth until school entry who are either impacted by or at risk of developing physical, intellectual, or mental disabilities. A significant portion of these children exhibit emotional or behavioral challenges, thus increasing the likelihood of displaying psychopathological symptoms, which in turn increases variance in the sample.

We hypothesized a significant correlation between parent-reported alexithymia levels in preschool-aged children and the alexithymia levels reported by their parents. Furthermore, we anticipated associations between both parental and child alexithymia and parent-reported psychopathological symptoms in children. Identifying these associations would underscore the pivotal role of early emotional comprehension in children's mental health and provide valuable insights for developing targeted early intervention strategies.

## Materials and methods

The study was approved by the Ethics Committee of the University of Regensburg (ID: 22-2878-101) and preregistered in the German Clinical Trials Register (DRKS; DRKS00029046). Participants were digitally provided with information on the study purpose, inclusion and exclusion criteria, and anonymous data collection and storage. Following this information, participants provided digital, informed consent by specifically ticking boxes indicating their agreement to participate, their agreement to anonymous data evaluation and that they are aware of the voluntary nature of the study. The recruitment period for this study spanned from 25/05/2022 to 15/01/2023.

### Participants

The desired sample size was determined via an a-priori power analysis in G*Power [21] for a multiple linear regression predicting children's psychopathology from 10 predictors. The correlation with alexithymia is reported at .62 for both anxiety and depression by Weissman et al. [13]. Brown et al. [14] found correlations between parent-reported alexithymia and psychopathology (measured via the Strengths and Difficulties Questionnaire), ranging between .23 and .30. Based on these effect sizes, the association of alexithymia with psychopathology is estimated to be of approximately medium size. To detect a medium effect of $R^2 = .13$ [22] at a

power of 80.0% in a multiple linear regression with ten predictors and α = 5%, a sample of 118 participants is necessary.

A total of 227 parents were recruited via local preschools and interdisciplinary intervention centers ("Frühförderstellen") in northeastern Bavaria, Germany. Out of these, $n = 53$ terminated the survey prematurely, resulting in a final sample of $N = 174$. Inclusion criteria were being the primary caregiver of a child between 3 and 6 years of age, children's regular preschool attendance, the absence of intellectual disability, and adequate comprehension of the German language. At the time of the survey, children had been attending preschool for an average of 1.22 years ($SD = 0.97$) and were on average 4.54 years old ($SD = 0.97$; 50.60% female). About 7.00% were bilingual and 66.10% had siblings. Interdisciplinary intervention program participation applied to 15.50% and 10.30% had speech problems (language delays were noted during the last routine examination by a medical doctor). Parents were between 22 and 44 years of age ($M = 34.48$, $SD = 4.82$; 75.30% female) and the majority was married (80.50%). The level of education was used as an approximation of socio-economic status and 31.60% had a university degree, 21.80% completed a-levels, 23.00% completed training school (Berufsschule), 13.20% completed 10 years of school education (Realschule), 9.80% completed 9 years of school education (Hauptschule), and 0.60% did not finish school. All parents had German nationality and spoke fluent German.

## Procedure

Flyers and postings were distributed at preschools and interdisciplinary intervention centers in northeastern Bavaria, Germany, including detailed information about the study and a QR code that parents could scan to access the online survey, which was programmed in LimeSurvey (Version 3.17.0, http://limesurvey.org). At the beginning of the survey, participants were briefed on the study's purpose and assured of voluntary participation, with the option to stop at any time or fully withdraw, resulting in the automatic deletion of their data. Details on anonymous data collection and storage were provided. Parents explicitly confirmed their first-time participation, their child's non-inclusion in previous study reports, their child's attendance at a standard preschool during data collection, and the absence of an intellectual disability diagnosis. Subsequently, they provided informed consent to participate. Initially, demographic information regarding the parent (age, sex, nationality, education) and the child (age, sex, participation in an IIP, speech problems) was collected. Following this, parents completed questionnaires on alexithymia, emotion recognition, and child psychopathology. Finally, they were directed to a separate survey where they had the option to provide their email address for compensation in the form of a 10-euro voucher. Participants were assured that their responses could not be linked to the provided email address.

## Material

**Perth Alexithymia Questionnaire (PAQ).** Questionnaire measures currently stand as the most widespread and reliable assessments of alexithymia across all age groups. The PAQ is a recently developed self-report measure designed to assess alexithymic traits in adults while addressing shortcomings of previous questionnaires by incorporating separate items for negative and positive feelings [23]. Participants rate 24 items on a 7-point Likert scale and subscales include "externally oriented thinking" (EOT) for attention to one's own feelings, "difficulty identifying feelings" (DIF), and "difficulty describing feelings" (DDF) for negative and positive feelings, respectively. Multiple composite scores can be derived, including a total alexithymia score and general difficulties in appraisal. The PAQ demonstrates good internal consistency (Cronbach's α = .87 to .96) and confirms its proposed factor structure [23, 24]. In this study,

the German version by Kaemmerer et al. [25] was utilized, showing high reliability in the present sample of parents (Cronbach's α = .97). The PAQ was chosen for its strong psychometric properties and its ability to differentiate between negative and positive feelings, setting it apart from other widely used alexithymia measures.

The German version of the PAQ [25] was adapted into a self-report version for children (PAQ-C) by Jarvers et al. [26], with adjustments made to language for child appropriateness. The PAQ-C retains the original item sequence and scoring instructions from the adult version. Validation for the PAQ-C is ongoing (Jarvers et al., 2021). In this study, the PAQ-C items were rephrased into third person perspective to create a parent-report version (PAQ-P). Cronbach's α for the total score was .96 in this study, with α ranging between .83 and .94 for subscales.

**Strengths and Difficulties Questionnaire (SDQ).** The SDQ is a widely used screening tool for assessing emotional and behavioral difficulties in children and adolescents [27]. It comprises 25 items rated on a 3-point scale. The instrument includes five subscales: emotional problems, conduct problems, hyperactivity, peer problems, and prosocial behavior. A total difficulties score is obtained by summing all subscale scores except prosocial behavior. The emotional problems and peer problems scales can be combined into an "internalizing" scale, while the conduct problems and hyperactivity scales can form an "externalizing" scale [20]. Internalizing refers to problems directed towards the inside (such as anxiety, depression, somatic complaints) and externalizing refers to problems directed towards the outside (such as aggression, hyperactivity, and oppositional defiance) [28]. For both scales values above 7 are considered at risk and values above 9 are considered critical. The SDQ demonstrates good internal consistency and discriminates well between clinical and non-clinical groups [29]. In this study, appropriate versions were administered automatically via LimeSurvey, with age-specific adaptations applied.

**The Reading the Mind in the Eyes Test (RMET).** The Reading the Mind in the Eyes Test (RMET) was initially developed as a measure of Theory of Mind (ToM) and participants are presented with photographs of eye regions and asked to infer mental and emotional states [30]. Although widely used in research on social cognition, recent discussions suggest it may primarily assess facial emotion recognition rather than ToM [31]. The RMET comprises 36 test items and one practice item, each with four forced-choice answer options. Test scores are calculated based on the number of correctly identified items. The original version has demonstrated some psychometric weaknesses, leading to the development of shorter forms. This study utilized a 10-item short version developed by Olderbak et al. [32], which showed improved homogeneity compared to the full version.

## Statistical analysis

Statistical analyses were conducted using the R statistical package, version 4.0.2 [33] and SPSS 28 [34]. Initially, correlations between children's and parental alexithymia were examined. Next, the impact of demographic variables such as age, sex, education, IIP participation, and speech problems on children's alexithymia and internalizing and externalizing psychopathology was assessed using Mann-Whitney $U$ tests and bivariate correlations. Correlational analyses were also employed to investigate whether specific subscores of child alexithymia were relevant to internalizing and externalizing symptomatology. Finally, linear multivariable regressions were conducted to identify significant predictors for a) children's alexithymia and b) children's internalizing and externalizing symptomatology. The predictors included in each regression model were parental alexithymia, parental emotion recognition, and parental care hours per day, considering that parents' reports may be influenced by the amount of time spent with their child. Multiple comparisons were controlled for using the False Discovery Rate (FDR, [35]) and alpha-level was set to 0.05.

**Table 1. Sample characteristics.**

| Variable | IIP Sample M (SD) | Non IIP Sample M (SD) | Total Sample M (SD) | Range |
|---|---|---|---|---|
| *Parent* | | | | |
| Age | 35.44 (5.20) | 34.31 (4.76) | 34.48 (4.82) | 22–44 |
| Care time (hours per day) | 9.54 (4.47) | 8.66 (4.58) | 8.78 (4.55) | 2–20 |
| Alexithymia (PAQ) | 54.07 (30.83) | 57.01 (30.23) | 56.65 (30.19) | 24–150 |
| Emotion recognition (RMET) | 7.11 (1.48) | 4.60 (2.54) | 5.02 (2.58) | 0–10 |
| *Child* | | | | |
| Age | 4.85 (0.91) | 4.51 (0.96) | 4.55 (0.96) | 3–6 |
| Alexithymia (PAQ-P) | 72.48 (31.45) | 72.29 (25.40) | 72.05 (26.51) | 24–132 |
| Internalizing (SDQ) | 8.00 (3.48) | 5.74 (2.94) | 6.06 (3.15) | 0–15 |
| Externalizing (SDQ) | 10.18 (4.01) | 6.42 (3.41) | 7.03 (3.74) | 0–18 |

*Note.* PAQ = Perth Alexithymia Questionnaire, RMET = Reading the Mind in the Eyes Test, PAQ-P = Perth Alexithymia Questionnaire Parent Report, SDQ = Strengths and Difficulties Questionnaire.

## Results

An overview of scores for parental and child variables across both groups (IIP and no IIP) and for the total sample is depicted in Table 1. Group differences between children attending an IIP and those who did not were present in parental education ($U = 1455.00$, $p = .026$, $r = .13$), internalizing and externalizing symptomatology ($U = 961.00–1202.50$, $p < .001$, $r = .24 - .32$), and parental emotion recognition skills ($U = 838.50$, $p < .001$, $r = .35$). Overall, children attending an IIP showed more psychopathology and had parents with lower education and better emotion recognition skills. All other variables were non-significant ($p > .05$). Across both groups, 16.7% of children were at risk and 25.4% of children had critical scores in internalizing psychopathology. For externalizing psychopathology, 22.5% of children were at risk and 38.2% had critical scores. There was a significant positive correlation between children's and parents' alexithymia total scores ($r = .37$, $p < .001$). The same applied to subscores ($p < .001$). See Fig 1 for a graphical depiction of alexithymia scores.

### Demographic variables

Children's alexithymia scores (measured by the PAQ-P) did not differ significantly between males and females ($U = 3512.00$, $p = .755$, $r = .02$), whether the children participated in an IIP ($U = 1881.00$, $p = .877$, $r = .01$), or whether they had speech problems ($U = 1155.00$, $p = .281$, $r = .08$). However, fathers reported significantly higher alexithymia scores for their children ($M = 82.00$, $SD = 18.7$) compared to mothers' reports ($M = 68.80$, $SD = 27.90$; $U = 1926.00$, $p = .006$, $r = .21$). Correlational analyses revealed a significant negative correlation between child alexithymia and parental age ($r = -.19$, $p < .001$) and a significant positive correlation with parental education ($r = .11$, $p = .048$).

Children's internalizing scores (measured by the SDQ) did not differ significantly between boys and girls ($U = 3661.00$, $p = .806$, $r = .02$) or based on the sex of the parent ($U = 2704.0$, $p =$

## Child and parental alexithymia scores

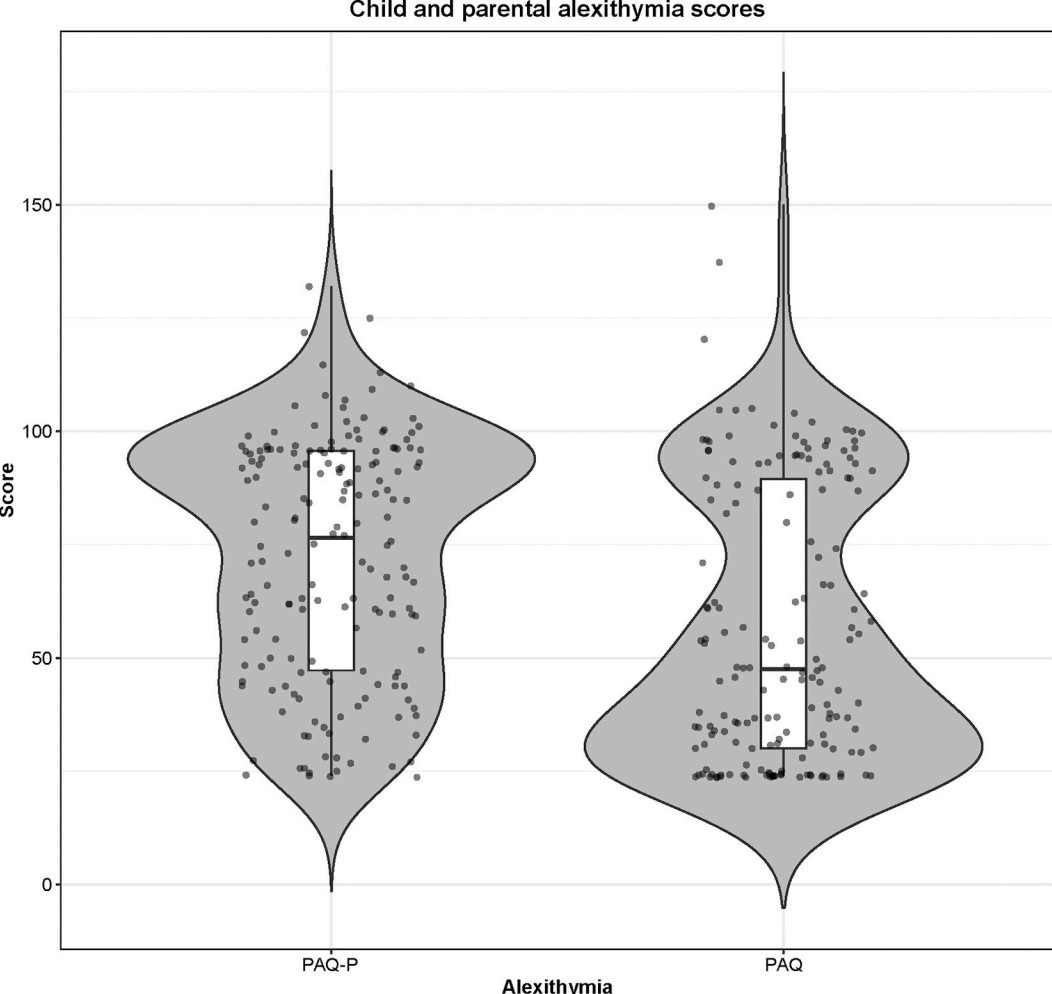

**Fig 1. Overview of child and parental alexithymia total score.** PAQ-P = Perth Alexithymia Questionnaire Parent Report, PAQ = Perth Alexithymia Questionnaire.

.747, $r = .02$). However, children with speech problems ($U = 823.0$, $p = .004$, $r = .22$; Speech$_{yes}$: $M = 8.1$, $SD = 2.9$, Speech$_{no}$: $M = 5.8$, $SD = 3.1$) and those participating in an IIP ($U = 1202.0$, $p = .001$, $r = .24$, see Table 1) had significantly higher internalizing scores compared to those who did not. Correlational analyses revealed a significant positive correlation of children's internalizing scores with child age ($r = .24$, $p < .001$) and a significant negative correlation with parental age ($r = -.15$, $p = .006$).

Children's externalizing scores (measured by the SDQ) did not differ significantly based on the sex of the parent ($U = 2338.0$, $p = .106$, $r = .12$). However, significant differences were found based on child sex ($U = 2825.0$, $p = .005$, $r = .21$; Boys: $M = 7.8$, $SD = 3.8$; Girls: $M = 6.3$, $SD = 3.6$), IIP participation ($U = 961.0$, $p < .001$, $r = .32$, see Table 1) and speech problems ($U = 722.5$, $p < .001$, $r = .26$; Speech$_{yes}$: $M = 9.8$, $SD = 3.7$; Speech$_{no}$: M = 6.7, SD = 3.6). Overall, boys, children participating in IIP, and children with speech problems showed higher externalizing scores. Parental age ($r = -.22$, p < .001) and parental education (r = - .14, p = .015) were significantly negatively correlated with externalizing scores.

**Table 2. Results of the linear regression model predicting children's alexithymia.**

| Dependent Variable | Predictor | *B* | SE | β | *t* | *p* | R² |
|---|---|---|---|---|---|---|---|
| Alexithymia (child) | Sex (child) | -1.93 | 3.51 | -0.04 | -0.55 | 0.583 | 0.34 |
| | Sex (parent) | -0.37 | 0.40 | -0.07 | -0.92 | 0.359 | |
| | Age (parent) | 0.45 | 4.41 | 0.01 | 0.10 | 0.918 | |
| | Education (parent) | 2.47 | 1.36 | 0.13 | 1.82 | 0.071 | |
| | **Alexithymia (parent)** | **0.38** | **0.07** | **0.44** | **5.77** | **<0.001** | |
| | **Emotion recognition (parent)** | **-2.01** | **0.77** | **-0.20** | **-2.60** | **0.010** | |
| | Care hours per day (parent) | -0.43 | 0.40 | -0.07 | -1.07 | 0.286 | |

*Note.* Alexithymia was measured using the Perth Alexithymia Questionnaire (original version for parents, parent report version for children). Emotion recognition in parents was assessed using the Reading the Mind in the Eyes Test.

### Predicting preschool children's alexithymia

To predict preschool children's alexithymia scores, a linear multivariable regression model was computed that incorporated demographic variables such as child sex, parent sex, parental age, and parental education. The model was significant ($F(7, 169) = 11.84$, $p < .001$) and accounted for 33.80% of the variance in children's alexithymia. The details of the predictors in this regression model are presented in Table 2.

### Predicting externalizing and internalizing symptomatology

Two linear multivariable regression models were computed to predict children's internalizing and externalizing symptomatology. Correlational analyses revealed significant correlations between all subscores of children's alexithymia and internalizing symptomatology (all $p < .002$), leading to the inclusion of the total alexithymia score in the model for internalizing symptoms. For externalizing symptomatology, the strongest association was found with difficulties describing negative feelings ($r = .23$, $p < .001$), which was included as a predictor. Additional demographic variables included child age, child sex, parental age, parental education, participation in an IIP, and speech problems.

The model predicting internalizing symptomatology was significant ($F(10,159) = 10.57$, $p < .001$) and accounted for 39.90% of the variance. Similarly, the model predicting externalizing symptomatology was significant ($F(10,159) = 12.79$, $p < .001$) and explained 44.60% of the variance. Detailed information on both models is provided in Table 3. Exploratory analyses using the alexithymia total score as predictor for externalizing symptomatology did not yield child alexithymia as a significant predictor.

Both regression analyses were repeated excluding the subgroup of children attending an IIP, and the results consistently showed the same pattern.

## Discussion

The present study aimed to investigate the relation between alexithymia in preschool children and their internalizing and externalizing symptomatology. Furthermore, parents' characteristics were considered as they are the main informants when studying early childhood and have a great influence on development during preschool age. We hypothesized a significant correlation between parent-reported alexithymia levels in preschool-aged children and the alexithymia levels reported by their parents. Additionally, we expected to observe links between both parental and child alexithymia and parent-reported internalizing and externalizing symptomatology in children.

**Table 3. Results of linear regression models predicting children's internalizing and externalizing symptomatology.**

| Dependent Variable | Predictor | *B* | SE | β | *t* | *p* | *R²* |
|---|---|---|---|---|---|---|---|
| Internalizing symptoms (child) | Sex (child) | 0.62 | 0.41 | 0.10 | 1.53 | 0.127 | 0.40 |
| | **Age (child)** | **0.99** | **0.21** | **0.30** | **4.64** | **<0.001** | |
| | **Alexithymia (child)** | **0.02** | **0.01** | **0.15** | **2.00** | **0.047** | |
| | **Intervention program** | **-1.54** | **0.59** | **-0.18** | **-2.60** | **0.010** | |
| | **Speech problems** | **1.82** | **0.72** | **0.18** | **2.54** | **0.012** | |
| | Age (parent) | -0.03 | 0.05 | -0.05 | -0.72 | 0.473 | |
| | Education (parent) | 0.00 | 0.16 | 0.00 | 0.01 | 0.993 | |
| | **Alexithymia (parent)** | **0.03** | **0.01** | **0.31** | **3.91** | **<0.001** | |
| | Emotion recognition (parent) | 0.03 | 0.09 | 0.02 | 0.31 | 0.758 | |
| | **Care hours per day (parent)** | **0.09** | **0.05** | **0.14** | **2.04** | **0.043** | |
| Externalizing symptoms (child) | Sex (child) | -0.54 | 0.47 | -0.07 | -1.17 | 0.245 | .45 |
| | **Age (child)** | **0.58** | **0.24** | **0.15** | **2.40** | **0.018** | |
| | **Alexithymia (child)** | **0.05** | **0.03** | **0.14** | **2.04** | **0.044** | |
| | **Intervention program** | **-1.82** | **0.68** | **-0.18** | **-2.67** | **0.008** | |
| | Speech problems | 1.22 | 0.82 | 0.10 | 1.48 | 0.141 | |
| | **Age (parent)** | **-0.12** | **0.05** | **-0.16** | **-2.26** | **0.025** | |
| | Education (parent) | -0.30 | 0.18 | -0.11 | -1.64 | 0.104 | |
| | **Alexithymia (parent)** | **0.04** | **0.01** | **0.31** | **4.15** | **<0.001** | |
| | **Emotion recognition (parent)** | **0.36** | **0.10** | **0.24** | **3.46** | **0.001** | |
| | Care hours per day (parent) | 0.10 | 0.05 | 0.12 | 1.96 | 0.052 | |

*Note.* SDQ (= Strengths and Difficulties Questionnaire) scales were used as outcome variables. Alexithymia was measured using the Perth Alexithymia Questionnaire (original version for parents, parent report version for children). For externalizing symptoms, the difficulties describing negative feelings scale is used. Emotion recognition in parents was assessed using the Reading Mind in the Eye Test.

## Preschool children's alexithymia

Most previous studies have not included children of preschool age or have examined young children only in combined samples with older children [13, 36]. Therefore, findings specifically on alexithymia in preschoolers constitute a novelty and illuminate the presentation of alexithymia in early childhood.

As expected, no significant sex differences were observed in alexithymia levels among preschoolers. While higher levels for men compared to women are discussed in adults [37], no theoretical basis suggests sex differences in early childhood. Factors highlighted by Levant et al. [37], i.e., socialization and expectations for men, may become relevant later in life, thus not yet impacting alexithymia levels in early childhood.

No significant association was found between the age of the children and their alexithymia levels. Karukivi and Saarijärvi [38] hypothesized that high alexithymia might be common, particularly in young children, given the ongoing development of emotional skills. However, the present sample was homogeneous regarding the respective life stage of the children, namely preschool. Thus, children may have been experiencing similar challenges in emotional development.

Surprisingly, no significant differences were found in alexithymia between children with and without speech problems. This finding contradicts the assumption that impaired language development plays a significant role in the etiology of alexithymia [39]. Possible explanations could be that the repercussions of language delays might manifest later in life, and children with identified speech delays may already be undergoing interventions.

Interestingly, fathers reported higher levels of alexithymia for their children compared to mothers. While family structures might vary between these informant groups, a reporting bias between fathers and mothers seems more plausible than genuine pronounced differences. Exploring relationship status and daily caregiving hours in relation to the sex of the reporting parent could be valuable in future analyses to clarify this group disparity.

When examining how much variance in children's alexithymia could be explained by demographic variables, as well as parental alexithymia and emotion recognition, only the latter two factors emerged as significant predictors. Parental alexithymia has been found to correlate with child alexithymia in previous studies [40]. This association may stem from various factors. Firstly, alexithymia has a genetic component, with an estimated heritability of around 30% [41], suggesting a positive correlation between parental and child alexithymia due to shared genetics alone. Furthermore, parenting style, known to be associated with parental alexithymia [42], may mediate the relationship between parental and child alexithymia, warranting further investigation in future studies. Parental emotion recognition likely predicts child alexithymia levels, but it may also affect how parents report on their child's alexithymia. Parents with lower emotion recognition skills might find it harder to accurately report on their children's emotional experiences, potentially biasing their responses.

## Alexithymia in relation to internalizing problems

In the present study, children's older age, higher alexithymia, participation in an IIP, speech problems and higher parental alexithymia as well as longer care hours were significant predictors for internalizing symptomatology. Given that data collection occurred in 2022, it is noteworthy to consider the potential influence of the ongoing COVID-19 pandemic on children's behavior, particularly for older children. In a study by Jarvers et al. [43], which surveyed behavior in preschoolers before, during, and after nationwide lockdowns in Germany, difficulties increased significantly during lockdown measures and persisted afterward. In the current study, conducted approximately a year later, it is possible that older children were more affected by lockdown measures and continue to exhibit more internalizing behavior as a consequence.

No sex difference could be detected regarding internalizing symptomatology, which is in line with previous findings [4]. Similarly, the impact of participation in an IIP and speech problems aligns with prior work: Developmental and medical issues in children have been consistently linked to internalizing behavior [44]. Concerning speech problems, a bidirectional relationship between internalizing problems and language difficulties has been suggested, where problems in communication can cause social withdrawal and vice versa [45].

As expected, child alexithymia (PAQ-P scores) demonstrated significant predictive value, with higher alexithymia scores predicting internalizing behavior. This association has been consistently observed in numerous studies, albeit with older samples [13, 14, 36]. The present findings align with prior research, providing valuable insights into an age group not previously studied on this specific topic. With the inclusion of a preschool sample, the association between alexithymia and internalizing problems has now been demonstrated consistently from early childhood to adulthood.

The association between alexithymia and psychopathology can be examined through different lenses. Hemming et al. [11] delineate several contrasting perspectives on this matter. The vulnerability hypothesis proposes that alexithymia serves as a predisposing factor for the emergence of mental health issues. Conversely, the reactivity hypothesis suggests that alexithymia emerges as a consequence of pre-existing conditions. While the current study's cross-sectional design constrains its ability to definitively resolve this debate, the discovery of this association

in a preschool sample implies that alexithymia is not solely a delayed outcome of internalizing problems. This finding lends support to the vulnerability hypothesis over the reactivity hypothesis.

A further significant predictor of internalizing symptomatology in preschool children was parental alexithymia. Parents reporting higher levels of alexithymia also reported increased internalizing behavior in their children. This is in line with previous work in school-aged children that identified parental alexithymia as a significant predictor for psychopathology [46]. Parental mental health issues are known to increase the risk of similar problems in children [4]. Given that alexithymia is associated with anxiety and depressive disorders in adults [12], highly alexithymic parents may be more susceptible to mental health issues themselves, affecting their children. Additionally, parental alexithymia is linked to parenting styles [42], which in turn can affect children's internalizing difficulties [47]. However, data on parental mental health and parenting styles were not collected in this study, so any discussion of mediating effects remains speculative.

At first, it may seem surprising that longer care hours are associated with higher internalizing symptomatology in preschool children. However, internalizing problems are often overlooked [48], especially in younger children and prolonged time spent with the child might increase the likelihood of identifying symptoms.

## Alexithymia in relation to externalizing problems

In the present study, significant contributors to externalizing symptomatology included children's older age, higher levels of alexithymia (difficulties describing negative feelings), and participation in an IIP, as well as parents' younger age, higher alexithymia, and better emotion recognition skills. Similar to internalizing symptomatology, older children may have been more affected by the COVID-19 pandemic compared to younger children who entered preschool later. This also applies to participation in an IIP, as developmental delays have been linked to externalizing symptomatology [49]. Interestingly, child sex did not emerge as a significant predictor, despite previous studies indicating higher externalizing symptomatology in boys [50]. One possible explanation could be that sex differences in externalizing behaviors are often attributed to language delays [51], which were partially accounted for in the model through speech problems and the difficulties describing negative feelings scale of the alexithymia questionnaire.

In contrast to internalizing symptomatology, only the alexithymia facet "difficulties describing negative feelings" was predictive of more parent-reported externalizing symptoms. Challenges in articulating one's negative emotional states may correlate with feelings of helplessness and anger, potentially leading to overwhelm and externalizing behaviors [52]. Effective communication of emotions, particularly negative ones, is vital in preschoolers for fostering emotion regulation, strongly linked to externalizing issues in this age group [53]. This focus on negative emotions aligns with previous studies utilizing the PAQ [23, 24, 54].

Parental factors, such as younger age, also had an impact on children's externalizing symptoms in the current sample. Younger parents, who may feel less prepared for parenthood, might resort to harsher parenting due to feelings of being overwhelmed, thereby potentially leading to increased externalizing behaviors in preschoolers [55]. Alternatively, less experience in parenting might influence reporting, with younger parents possibly rating their children's behavior as more pathological compared to older parents.

Parental alexithymia also demonstrated an effect and may impact children's externalizing symptomatology in a similar fashion to internalizing symptomatology, notably through parenting styles and parental mental health issues [12, 42].

Finally, enhanced parental emotional recognition skills were linked with elevated externalizing symptomatology in preschool children. The direction of this relationship remains unclear due to the cross-sectional design of the present study; therefore, it is plausible that parents of children exhibiting externalizing symptoms have developed heightened sensitivity towards emotion expressions in order to respond promptly. Alternatively, there could be a reporting bias, wherein parents with superior emotion recognition skills tend to report more externalizing behaviors in their children.

## Strengths and limitations

The present study benefited from several significant strengths. Methodological decisions allowed for the examination of an understudied age group regarding alexithymia within a sizable sample exceeding statistical requirements. Additionally, the study controlled for various factors, including subjective (questionnaires) and objective measures (RMET). Furthermore, the utilization of the newly developed PAQ provided a comprehensive assessment of difficulties with positive and negative emotions, as well as excellent psychometric properties.

However, also some limitations of the current study need to be addressed to contextualize its findings. One concern is the representativeness of the sample under investigation. The participants comprised children aged 3 to 6 years attending regular preschools in northeastern Bavaria, Germany. While this might enhance the homogeneity of the sample, it may exclude children who are cared for at home. However, it's worth noting that in the year 2022, 91.7% of all 3- to 6-year-olds in Germany attended preschool [56], indicating that the majority of children would have been eligible for this study. Future research should investigate alexithymia in preschool-aged children who do not attend preschool. This is important because peer interactions increase during this period, and children interact more with alternative caregivers who may possess resources for emotional knowledge that children not attending preschool may lack [15]. As our sample consists of children in Germany, it is crucial to consider cultural aspects, particularly differences in emotional language and expression across cultures [57]. Nevertheless, previous research on alexithymia and psychopathology suggests similar relationships across countries [58]. An additional aspect that may impact the generalizability of the findings is the slight overrepresentation of individuals with a university degree in our sample. Preschool attendance can be costly, and not all parents in Germany are able to secure a spot for their child, making higher education levels likely in a preschool-attending sample [43].

Several limitations are inherent in the methodology of this study. Despite extensive plausibility checks, determining the accuracy of responses in an anonymous online survey presents a challenge and needs to be considered. Moreover, while the original version of the PAQ shows excellent reliability and validity across several languages [24, 54], and showed excellent reliability in our sample, the adapted version used to assess child alexithymia (the PAQ-P) lacks validation as it was adapted specifically for this study.

Another methodological limitation is that this study relied on only one rater (one parent) and solely utilized questionnaires. However, no reliable objective measures are available to assess alexithymia, no matter the age, and parents have been shown to be reliable informants on child psychopathology [19, 20], especially during preschool age [59]. Additionally, using parent-report questionnaires enhances comparability with many other studies, as research on children's emotions and behaviors frequently involves surveying only one parent [4, 14, 36].

Finally, our study was a cross-sectional investigation of alexithymia and psychopathology in preschool children. Although it is generally assumed that alexithymia develops early and precedes the onset of mental health problems [60], this study cannot establish causality and only examines the correlational relationship between these variables in children aged 3 to 6. Future

longitudinal research is needed to explore the development of alexithymia and psychopathology over time, enabling a better understanding of the directionality of their relationship.

## Conclusion

Overall, early childhood, specifically preschool age, has received limited attention in alexithymia research. The present study investigated the relation between alexithymia in preschool children and their internalizing and externalizing symptomatology, considering parental factors like parental alexithymia and parental emotion recognition skills. We found that both child alexithymia and parental alexithymia were significant predictors of internalizing and externalizing psychopathology in preschool children. To the best of our knowledge, this is the first study to explore alexithymia and related concepts exclusively within a preschool sample. These findings offer initial insights into the role of alexithymia as a risk factor in early childhood, providing a basis for future research aimed at understanding the causes, prevention, and intervention strategies for internalizing and externalizing difficulties in children.

## Acknowledgments

The authors would like to thank all parents and children who were part of this study. No funding was received for this study.

## Author Contributions

**Conceptualization:** Irina Jarvers, Eva Kormann, Daniel Schleicher, Angelika Ecker, Stephanie Kandsperger, Romuald Brunner.

**Data curation:** Irina Jarvers, Eva Kormann.

**Formal analysis:** Irina Jarvers.

**Investigation:** Irina Jarvers, Eva Kormann.

**Methodology:** Irina Jarvers, Eva Kormann, Daniel Schleicher, Angelika Ecker, Stephanie Kandsperger.

**Project administration:** Irina Jarvers.

**Software:** Irina Jarvers.

**Supervision:** Irina Jarvers, Romuald Brunner.

**Validation:** Eva Kormann, Daniel Schleicher, Angelika Ecker, Stephanie Kandsperger, Romuald Brunner.

**Visualization:** Irina Jarvers.

**Writing – original draft:** Irina Jarvers.

**Writing – review & editing:** Eva Kormann, Daniel Schleicher, Angelika Ecker, Stephanie Kandsperger, Romuald Brunner.

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
