## [Decision Letter · Decision Letter 0]

2 Jul 2024

PONE-D-24-15302Impact of alexithymia, speech problems and parental emotion recognition on internalizing and externalizing problems in preschoolersPLOS ONE

Dear Dr. Jarvers,

Thank you for submitting your manuscript to PLOS ONE. After careful consideration, we feel that it has merit but does not fully meet PLOS ONE’s publication criteria as it currently stands. Therefore, we invite you to submit a revised version of the manuscript that addresses the points raised during the review process.

We look forward to receiving your revised manuscript.

Kind regards,

Runtang Meng, PhD

Academic Editor

PLOS ONE

2. In the online submission form you indicate that your data is not available for proprietary reasons and have provided a contact point for accessing this data. Please note that your current contact point is a co-author on this manuscript. According to our Data Policy, the contact point must not be an author on the manuscript and must be an institutional contact, ideally not an individual. Please revise your data statement to a non-author institutional point of contact, such as a data access or ethics committee, and send this to us via return email. Please also include contact information for the third party organization, and please include the full citation of where the data can be found.

Additional Editor Comments (if provided):

Reviewers' comments:

Reviewer's Responses to Questions

**Comments to the Author**

1. Is the manuscript technically sound, and do the data support the conclusions?

Reviewer #1: Yes

Reviewer #2: Yes

Reviewer #3: Yes

Reviewer #4: No

Reviewer #5: Yes

Reviewer #6: No

2. Has the statistical analysis been performed appropriately and rigorously? 

Reviewer #1: Yes

Reviewer #2: Yes

Reviewer #3: Yes

Reviewer #4: N/A

Reviewer #5: Yes

Reviewer #6: Yes

3. Have the authors made all data underlying the findings in their manuscript fully available?

Reviewer #1: Yes

Reviewer #2: Yes

Reviewer #3: Yes

Reviewer #4: Yes

Reviewer #5: Yes

Reviewer #6: No

4. Is the manuscript presented in an intelligible fashion and written in standard English?

Reviewer #1: Yes

Reviewer #2: Yes

Reviewer #3: Yes

Reviewer #4: Yes

Reviewer #5: Yes

Reviewer #6: Yes

5. Review Comments to the Author

Reviewer #1: The article provides a compelling examination of alexithymia in preschool-aged children, an area that has been scarcely explored in the literature despite its significant implications for mental health. The study addresses a notable gap by investigating the relationship between child and parental alexithymia and its impact on early childhood psychopathology.

The research is well-written, presenting a clear methodology involving data from 174 parents of children aged 3 to 6. The use of validated questionnaires to assess alexithymia and psychopathology in both children and parents is a strong point, as it allows for a robust analysis of the relationships between these variables. The findings highlight the predictive power of parental alexithymia and emotion recognition abilities on child alexithymia, and how these factors relate to internalizing and externalizing symptomatology in children.

Notably, the study underscores the importance of alexithymia as a potential risk factor in early childhood, providing a foundation for future research aimed at understanding its causes, prevention, and intervention strategies for associated psychopathology. This perspective opens new avenues for exploring early intervention techniques and enhancing parental support programs to mitigate the development of mental health issues in children.

The article's limitations are punctually addressed, further strengthening its validity. The authors' thorough discussion of these limitations ensures that the conclusions drawn are well-supported and that the study provides a comprehensive overview of the topic.

Overall, this study is a valuable contribution to the field, offering fresh insights and setting the stage for future research on alexithymia and childhood psychopathology.

Reviewer #2: the article titled Impact of alexithymia, speech problems and parental emotion recognition on

internalizing and externalizing problems in preschoolers is an innovative topic and difficult to conduct for preschoolers too. It will be the foundational stone for further related studies.

Reviewer #3: The article presented by the authors aims to fill in the gaps on the topic of alaxithaemia, where research is scarce. The subject is complex, but the author has produced a very detailed and meticulous article. Previous articles have mainly looked at the adult or pre-school age, leaving out early childhood, which turns out to be extremely important and little studied, given the difficulty of the patient. In addition to congratulating the authors for having written an article on a very complex subject, providing data and considerations that are very important for the entire scientific community, I would like to make the following suggestions to improve an article that is already of a very high standard.

The objectives set by the authors have been respected and developed in an appropriate manner, providing an important starting point for future studies. First of all, the entire article should be reviewed in English, as there are redundant terms that can be improved by synonyms, if not grammatical errors that can be easily corrected.

Introduction:

38-44: I would pay more attention to this paragraph, as it is of particular importance for the purpose of the article, I would improve its drafting, including any studies, regarding the identification of "pathology" when caregivers struggle to identify and interpret the child's emotions, identifying possible and viable solutions to remedy this problem.

Study design:

The study design described in the introduction was explained in a clear and simple manner.

Materials and methods:

The flow of data collection, with the criteria used for the study, has been described in an appropriate and thorough manner, there are no corrections to be made as they are meticulous. The statistical data produced are meaningful, clear and easy to understand.

I would suggest an extension of the study court, for future implementation of the study, as it is often and scientifically important.

The material presented for data collection, was well documented, described it impeccably, there are no corrections to be made.

Statistical analysis, well documented, highlighting the strengths and limitations of the study.

Conclusions:

The article is very long, but the importance of the topic justifies the length. All data, results and discussions are well documented. The article is of high scientific importance, a necessary result for the whole scientific community, with the aim of increasing knowledge in a topic that is still little explored. All data and methods used are user-friendly. The authors have presented an article that adheres to appropriate reporting guidelines and community standards for data availability. Applicable standards for study ethics and research integrity are met. The manuscript is well written and I believe it can be accepted for publication. Implementation with a series of case study articles is necessary for the future development of the topic of scientific interest, as I believe it will benefit the entire scientific community.

Reviewer #4: 1. In the abstract of the original article, the details of the materials and methods should be presented separately from the results. This ensures clarity and facilitates understanding of both the procedures employed and the findings obtained.

2. In Table 2 and 3: Maintain consistency in the notation and format of numerical data. The use of zero as an integer and then omitting it elsewhere could cause confusion. Uniformity in the presentation of results.

3. In the final paragraph of the discussion, spaced 9-10, it is noted that authors are cited in a format different from the Vancouver style. Unify the citation style with the rest of the content.

4. The conclusion focuses more on the relevance of the study and its implications for future research, rather than detailing specific findings that address the purpose of the study.

Reviewer #5: Interesting study from the point of view of the mental health of parents and children, however there are some observations that I detail below:

Summary: It is missing to place the study design and briefly the selection criteria. The wording of the statistics used is not understood.

Introduction: Improve the wording by being more precise with what has been found in the review of the scientific evidence, improve the wording of the justification, and should conclude with the objective of the study.

Material and method: Adequately describe the selection criteria, adequately detail the procedural phase, be more precise with the wording of the statistical analysis.

Results: Improve the wording of the interpretation of the results.

Conclusion: Be more precise and objective in the wording.

Reviewer #6: The study presents an important and under-researched area in understanding alexithymia in early childhood and its implications for psychopathology, providing valuable insights into the potential intergenerational transmission of emotional processing difficulties. While the topic is relevant and the research question is well-framed, there are several methodological and interpretative concerns that need to be addressed to strengthen the manuscript.

Major Comments:

Sample Size and Generalizability:

The sample size of 174 parents, including a subgroup of 27 children in an intervention program, raises concerns about the generalizability of the findings. More information on how these participants were selected and how representative they are of the broader population is necessary.

Consider discussing the limitations of the sample size more explicitly and how it might impact the study's conclusions.

Measurement Tools:

The reliance on parent-reported measures (Perth Alexithymia Questionnaire – Parent Report and Strengths and Difficulties Questionnaire) introduces a potential bias. Parents’ perceptions may not accurately reflect the child’s true emotional state or behavior.

It would strengthen the study to include direct assessments of the children, possibly through observational methods or clinician-administered measures.

Intervention Program Group:

The inclusion of 27 children in an interdisciplinary intervention program could introduce confounding variables. The differences between this subgroup and the rest of the sample should be clarified.

An analysis comparing outcomes between children in the intervention program and those not in the program would be beneficial.

Causality and Directionality:

The cross-sectional nature of the study limits the ability to draw causal conclusions. While the study suggests correlations between parental alexithymia, child alexithymia, and psychopathology, it cannot establish causality.

Consider discussing longitudinal designs or other methods that could be used in future research to better understand the causal relationships.

Minor Comments:

Literature Review:

The introduction could benefit from a more thorough review of existing literature on alexithymia in early childhood and its links to psychopathology to provide a stronger foundation for the study.

Clarification of Terms:

Terms such as "internalizing symptomatology" and "externalizing symptomatology" should be clearly defined for readers who may not be familiar with these concepts.

Formatting and Writing:

Ensure consistency in the use of terms and abbreviations throughout the manuscript.

Minor grammatical and typographical errors should be corrected to improve readability.

Conclusion:

While the study addresses a significant gap in the literature and provides preliminary evidence on the role of alexithymia in early childhood, the methodological and interpretative concerns need to be addressed to ensure the findings are robust and generalizable. Further research with larger, more representative samples and longitudinal designs is recommended to build on these initial findings.

6. PLOS authors have the option to publish the peer review history of their article (what does this mean?). If published, this will include your full peer review and any attached files.

Reviewer #1: No

Reviewer #2: No

Reviewer #3: No

Reviewer #4: **Yes: **Jose Luis Huamani-Echaccaya

Reviewer #5: No

Reviewer #6: No

---

## [Author Response · Author response to Decision Letter 0]

8 Jul 2024

Reviewer #1: 

The article provides a compelling examination of alexithymia in preschool-aged children, an area that has been scarcely explored in the literature despite its significant implications for mental health. The study addresses a notable gap by investigating the relationship between child and parental alexithymia and its impact on early childhood psychopathology.

The research is well-written, presenting a clear methodology involving data from 174 parents of children aged 3 to 6. The use of validated questionnaires to assess alexithymia and psychopathology in both children and parents is a strong point, as it allows for a robust analysis of the relationships between these variables. The findings highlight the predictive power of parental alexithymia and emotion recognition abilities on child alexithymia, and how these factors relate to internalizing and externalizing symptomatology in children.

Notably, the study underscores the importance of alexithymia as a potential risk factor in early childhood, providing a foundation for future research aimed at understanding its causes, prevention, and intervention strategies for associated psychopathology. This perspective opens new avenues for exploring early intervention techniques and enhancing parental support programs to mitigate the development of mental health issues in children.

The article's limitations are punctually addressed, further strengthening its validity. The authors' thorough discussion of these limitations ensures that the conclusions drawn are well-supported and that the study provides a comprehensive overview of the topic.

Overall, this study is a valuable contribution to the field, offering fresh insights and setting the stage for future research on alexithymia and childhood psychopathology.

ANSWER:

Dear Reviewer 1,

Thank you for your thoughtful and encouraging review of our manuscript. We are delighted that you found our study to be a compelling examination of alexithymia in preschool-aged children and its significant implications for mental health. Your comments on the importance of alexithymia as a potential risk factor in early childhood and the need for future research on its causes, prevention, and intervention strategies are particularly encouraging. 

Sincerely,

The authors

Reviewer #2:

the article titled Impact of alexithymia, speech problems and parental emotion recognition on

internalizing and externalizing problems in preschoolers is an innovative topic and difficult to conduct for preschoolers too. It will be the foundational stone for further related studies.

ANSWER:

Dear Reviewer 2,

Thank you for your positive feedback on our manuscript titled “Impact of Alexithymia, Speech Problems, and Parental Emotion Recognition on Internalizing and Externalizing Problems in Preschoolers." We are pleased that you found the topic innovative and recognized the challenges of conducting such research with preschoolers.

Your acknowledgment of our study as a foundational stone for future related studies is greatly appreciated. We hope our work will inspire and facilitate further research in this important area.

Sincerely,

The authors

Reviewer #3: The article presented by the authors aims to fill in the gaps on the topic of alexithymia, where research is scarce. The subject is complex, but the author has produced a very detailed and meticulous article. Previous articles have mainly looked at the adult or pre-school age, leaving out early childhood, which turns out to be extremely important and little studied, given the difficulty of the patient. In addition to congratulating the authors for having written an article on a very complex subject, providing data and considerations that are very important for the entire scientific community, I would like to make the following suggestions to improve an article that is already of a very high standard.

The objectives set by the authors have been respected and developed in an appropriate manner, providing an important starting point for future studies. First of all, the entire article should be reviewed in English, as there are redundant terms that can be improved by synonyms, if not grammatical errors that can be easily corrected.

ANSWER:

Dear Reviewer 3,

Thank you very much for the thorough and encouraging review of our manuscript titled “Impact of Alexithymia, Speech Problems, and Parental Emotion Recognition on Internalizing and Externalizing Problems in Preschoolers”, submitted for publication in PLOS ONE. We have thoroughly reviewed the manuscript to address grammatical errors and enhance readability by refining the use of synonyms where appropriate. We are very grateful for your helpful suggestions. Please find a detailed point-by-point response below. The respective changes in the manuscript are highlighted with track-changes to facilitate the follow-up. Page and line numbers refer to the track-changes document with changes visible.

Sincerely,

The authors

Introduction:

38-44: I would pay more attention to this paragraph, as it is of particular importance for the purpose of the article, I would improve its drafting, including any studies, regarding the identification of "pathology" when caregivers struggle to identify and interpret the child's emotions, identifying possible and viable solutions to remedy this problem.

ANSWER: Thank you very much for this important suggestion. Parent-reports on children’s psychopathology, especially in preschool children, are reported to be stable and reliable with few effects of confounding variables. Unfortunately, to our knowledge, there has only been a single study investigating the impact of parental alexithymia on parental report of children’s emotional skills. Thus, we have elaborated on the conducted study and emphasized its implications. As suggested, we have pointed out that assessing these variables is a possible solution to control for their impact (page 5, lines 94 – 105).

Study design:

The study design described in the introduction was explained in a clear and simple manner.

ANSWER:Thank you very much.

Materials and methods:

The flow of data collection, with the criteria used for the study, has been described in an appropriate and thorough manner, there are no corrections to be made as they are meticulous. The statistical data produced are meaningful, clear and easy to understand.

ANSWER:Thank you for your appreciation.

I would suggest an extension of the study court, for future implementation of the study, as it is often and scientifically important.

ANSWER:Thank you very much for pointing this out. The purpose of the present study was to gain a first glance at alexithymia in preschool age in a sample with sufficient power for the conducted analyses. Nevertheless, we fully agree that an extension of the sample size would benefit our conclusions and are planning a future follow up study with more objective measures of child alexithymia and a larger sample size. 

The material presented for data collection, was well documented, described it impeccably, there are no corrections to be made.

ANSWER:Thank you very much.

Statistical analysis, well documented, highlighting the strengths and limitations of the study.

ANSWER:We are grateful for the positive feedback.

Conclusions:

The article is very long, but the importance of the topic justifies the length. All data, results and discussions are well documented. The article is of high scientific importance, a necessary result for the whole scientific community, with the aim of increasing knowledge in a topic that is still little explored. All data and methods used are user-friendly. The authors have presented an article that adheres to appropriate reporting guidelines and community standards for data availability. Applicable standards for study ethics and research integrity are met. The manuscript is well written and I believe it can be accepted for publication. Implementation with a series of case study articles is necessary for the future development of the topic of scientific interest, as I believe it will benefit the entire scientific community.

ANSWER:Thank you for your positive evaluation of our article. We appreciate your recognition of the importance and thoroughness of our work, and we are pleased that you find the manuscript suitable for publication. We will certainly consider your suggestion for future studies to further explore this important topic.

Reviewer #4: 

ANSWER:

Dear Reviewer 4,

Thank you very much for the thorough and helpful review of our manuscript titled “Impact of Alexithymia, Speech Problems, and Parental Emotion Recognition on Internalizing and Externalizing Problems in Preschoolers”, submitted for publication in PLOS ONE. We have revised the manuscript to address your helpful suggestions. Please find a detailed point-by-point response below. The respective changes in the manuscript are highlighted with track-changes to facilitate the follow-up. Page and line numbers refer to the track-changes document with changes visible.

Sincerely,

The authors

1. In the abstract of the original article, the details of the materials and methods should be presented separately from the results. This ensures clarity and facilitates understanding of both the procedures employed and the findings obtained.

ANSWER:Thank you very much for this suggestion. We have gladly adjusted the abstract to include separate sections for methods and results (page 2, lines 26 – 32)

2. In Table 2 and 3: Maintain consistency in the notation and format of numerical data. The use of zero as an integer and then omitting it elsewhere could cause confusion. Uniformity in the presentation of results.

ANSWER:Thank you for this suggestion. In our tables we have followed the APA Style requirements which recommend removing leading zeros for values that cannot become larger than 0 (e.g., p-values). We have now adjusted the notation across tables, ensuring leading zeros are included throughout.

3. In the final paragraph of the discussion, spaced 9-10, it is noted that authors are cited in a format different from the Vancouver style. Unify the citation style with the rest of the content.

ANSWER:Thank you very much for spotting this. We have adjusted the citation format to ensure uniformity.

4. The conclusion focuses more on the relevance of the study and its implications for future research, rather than detailing specific findings that address the purpose of the study.

ANSWER:This is an important remark and we have adjusted the conclusion to include detailed findings and implications for future research as a follow-up (page 29, lines 536 – 550).

Reviewer #5: Interesting study from the point of view of the mental health of parents and children, however there are some observations that I detail below:

ANSWER:

Dear Reviewer 5,

Thank you very much for the thorough and helpful review of our manuscript titled “Impact of Alexithymia, Speech Problems, and Parental Emotion Recognition on Internalizing and Externalizing Problems in Preschoolers”, submitted for publication in PLOS ONE. We have revised the manuscript to address your helpful suggestions. Please find a detailed point-by-point response below. The respective changes in the manuscript are highlighted with track-changes to facilitate the follow-up. Page and line numbers refer to the track-changes document with changes visible.

Sincerely,

The authors

Summary: It is missing to place the study design and briefly the selection criteria. The wording of the statistics used is not understood.

ANSWER:Thank you very much for pointing this out. We have gladly incorporated information on the study design (online survey), the selection criteria (attending regular preschool) and included information on the statistical method used (multivariable linear regressions). Please see page 2, lines 26 – 34.

Introduction: Improve the wording by being more precise with what has been found in the review of the scientific evidence, improve the wording of the justification, and should conclude with the objective of the study.

ANSWER:We are grateful for the provided suggestions and have overworked the introduction to improve wording and formulations. We added additional sentences to point out the justification of the study (i.e., extensive research suggestion alexithymia as a risk factor but no research on alexithymia in preschool children; page 4, lines 77 – 84). The introduction now ends with explicit mention of the objective of the study and our hypotheses.

Material and method: Adequately describe the selection criteria, adequately detail the procedural phase, be more precise with the wording of the statistical analysis.

ANSWER:Thank you for this recommendation. We have extended our description of the selection criteria which were being the parent of children aged 3 to 5, the child attending a regular preschool, absence of intellectual disability, and adequate comprehension of the German language (page 7, lines 156 – 159). Additionally, we have elaborated on the procedural phase and recruitment (page 8, lines 171 – 175) and the statistical analysis (page 11, lines 242 – 256).

Results: Improve the wording of the interpretation of the results.

ANSWER:We have gladly implemented this recommendation and overworked the results, including more details and specifics in the descriptions (pages 12-16, lines 258 – 336).

Conclusion: Be more precise and objective in the wording.

ANSWER:Thank you for pointing this out. We have overworked the conclusion to involve more information on the objective results of the study with a brief outlook at possible implications and future research (page 29, lines 536 – 550).

Reviewer #6: The study presents an important and under-researched area in understanding alexithymia in early childhood and its implications for psychopathology, providing valuable insights into the potential intergenerational transmission of emotional processing difficulties. While the topic is relevant and the research question is well-framed, there are several methodological and interpretative concerns that need to be addressed to strengthen the manuscript.

ANSWER:

Dear Reviewer 6,

Thank you very much for the thorough and encouraging review of our manuscript titled “Impact of Alexithymia, Speech Problems, and Parental Emotion Recognition on Internalizing and Externalizing Problems in Preschoolers”, submitted for publication in PLOS ONE. We have revised the manuscript to address your helpful suggestions. Please find a detailed point-by-point response below. The respective changes in the manuscript are highlighted with track-changes to facilitate the follow-up. Page and line numbers refer to the track-changes document with changes visible.

Sincerely,

The authors

Major Comments:

Sample Size and Generalizability:

The sample size of 174 parents, including a subgroup of 27 children in an intervention program, raises concerns about the generalizability of the findings. More information on how these participants were selected and how representative they are of the broader population is necessary. Consider discussing the limitations of the sample size more explicitly and how it might impact the study's conclusions.

ANSWER:Thank you very much for pointing out this concern. Regarding the sample size itself we have conducted a power-analysis and determined that our sample size supersedes the 118 participants necessary to be able to draw meaningful statistical conclusions. However, our sample constitutes children in regular preschools in northeastern Bavaria, Germany, and generalizability to samples that do not attend preschool or with different cultural backgrounds may be difficult. We have added information on the selection of participants in the methods section and pointed out that all children attending regular preschools in northeastern Bavaria, Germany, were able to participate (page 8, lines 171 – 175) and included a discussion of the generalizability in the limitations (page 27, line 499 – 506). Additionally, we have added higher parental education as a point that may reduce generalizability in the limitations (page 28, lines 507 – 510). 

Measurement Tools:

The reliance on parent-reported measures (Perth Alexithymia Questionnaire

---

## [Decision Letter · Decision Letter 1]

28 Aug 2024

Impact of alexithymia, speech problems and parental emotion recognition on internalizing and externalizing problems in preschoolers

PONE-D-24-15302R1

Dear Dr. Jarvers,

We’re pleased to inform you that your manuscript has been judged scientifically suitable for publication and will be formally accepted for publication once it meets all outstanding technical requirements.

Kind regards,

Runtang Meng, PhD

Academic Editor

PLOS ONE

Additional Editor Comments (optional):

Reviewers' comments:

Reviewer's Responses to Questions

**Comments to the Author**

1. If the authors have adequately addressed your comments raised in a previous round of review and you feel that this manuscript is now acceptable for publication, you may indicate that here to bypass the “Comments to the Author” section, enter your conflict of interest statement in the “Confidential to Editor” section, and submit your "Accept" recommendation.

Reviewer #1: All comments have been addressed

Reviewer #2: All comments have been addressed

Reviewer #3: All comments have been addressed

Reviewer #5: All comments have been addressed

2. Is the manuscript technically sound, and do the data support the conclusions?

Reviewer #1: Yes

Reviewer #2: Yes

Reviewer #3: Yes

Reviewer #5: Yes

3. Has the statistical analysis been performed appropriately and rigorously? 

Reviewer #1: Yes

Reviewer #2: I Don't Know

Reviewer #3: Yes

Reviewer #5: Yes

4. Have the authors made all data underlying the findings in their manuscript fully available?

Reviewer #1: Yes

Reviewer #2: Yes

Reviewer #3: Yes

Reviewer #5: Yes

5. Is the manuscript presented in an intelligible fashion and written in standard English?

Reviewer #1: Yes

Reviewer #2: Yes

Reviewer #3: Yes

Reviewer #5: Yes

6. Review Comments to the Author

Reviewer #1: The revised methodology now provides a more comprehensive description of the study sample, including detailed inclusion and exclusion criteria. This addition strengthens the validity of the findings by ensuring that the sample is well-defined and appropriate for the research objectives. Furthermore, the procedures for measuring alexithymia, speech problems, and parental emotion recognition are now described with greater precision, including the specific tools and their psychometric properties. This thoroughness allows for better replication and understanding of the study design.

Overall, the changes inserted in the procedures and methods sections contribute to making this version of the article an even better one than the previous submission. The revisions provide greater transparency and rigor, which enhance the study's contribution to the existing literature on alexithymia, speech problems, and emotional development in preschoolers

Reviewer #2: the article titled Impact of alexithymia, speech problems and parental emotion recognition on

internalizing and externalizing problems in preschoolers is an innovative, complex and meticulous topic and difficult to conduct for preschoolers too. all the comments have been addressed but the statistics must be reviewed by an expert biostatistician.

Reviewer #3: The article presents a comprehensive and detailed account of the subject of atrial fibrillation, which is currently under-researched. Despite the complexity of the topic, the author has produced a highly detailed and meticulous article. The authors have met and developed their objectives in a way that sets an important starting point for future studies. Firstly, the article should be reviewed in English. This will remove redundant terms and improve the language where necessary. It will also resolve any grammatical errors.The study design was as follows:

The study design, described in the introduction, was explained in a clear and simple manner.

The materials and methods are clearly and accurately presented.

The data collection flow and the criteria used for the study were described in a suitable and thorough manner. There are no corrections to be made, as everything appears to be meticulous. The statistical data produced are significant, clear and easy to understand.

I propose extending the study court for future implementation of the study, as it is often and scientifically important.

The material presented for data collection was well documented, describing it impeccably. There are no corrections to be made.

In conclusion, it is clear that…

The article is comprehensive and deserves to be read in full, given the importance of the topic. All data, results and discussions are fully documented. This article is of great scientific importance and represents a significant contribution to the field of study. It addresses a topic that has been largely unexplored and is a valuable addition to the existing body of knowledge. The data and methods used are user-friendly. The authors have presented an article that adheres to appropriate reporting guidelines and community standards for data availability. They have also met the applicable standards for study ethics and research integrity. The manuscript is well written and should be accepted for publication. To further develop this topic of scientific interest, future work should include an implementation with a number of case study articles. This will benefit the entire scientific community.

Reviewer #5: It is an interesting topic that addresses a mental health issue related to parents and their children. It is well structured.

7. PLOS authors have the option to publish the peer review history of their article (what does this mean?). If published, this will include your full peer review and any attached files.

Reviewer #1: No

Reviewer #2: No

Reviewer #3: No

Reviewer #5: No

---

## [Editor Report · Acceptance letter]

2 Sep 2024

PONE-D-24-15302R1 

PLOS ONE

Dear Dr. Jarvers, 

I'm pleased to inform you that your manuscript has been deemed suitable for publication in PLOS ONE. Congratulations! Your manuscript is now being handed over to our production team.

Kind regards, 

on behalf of

Dr. Runtang Meng 

Academic Editor

PLOS ONE